# Synaptic up-scaling preserves motor circuit output after chronic, natural inactivity

Joseph M Santin[1,2]*, Mauricio Vallejo[2], Lynn K Hartzler[2]

[1]Division of Biological Sciences, University of Missouri-Columbia, Columbia, United States; [2]Department of Biological Sciences, Wright State University, Dayton, United States

**Abstract** Neural systems use homeostatic plasticity to maintain normal brain functions and to prevent abnormal activity. Surprisingly, homeostatic mechanisms that regulate circuit output have mainly been demonstrated during artificial and/or pathological perturbations. Natural, physiological scenarios that activate these stabilizing mechanisms in neural networks of mature animals remain elusive. To establish the extent to which a naturally inactive circuit engages mechanisms of homeostatic plasticity, we utilized the respiratory motor circuit in bullfrogs that normally remains inactive for several months during the winter. We found that inactive respiratory motoneurons exhibit a classic form of homeostatic plasticity, up-scaling of AMPA-glutamate receptors. Up-scaling increased the synaptic strength of respiratory motoneurons and acted to boost motor amplitude from the respiratory network following months of inactivity. Our results show that synaptic scaling sustains strength of the respiratory motor output following months of inactivity, thereby supporting a major neuroscience hypothesis in a normal context for an adult animal.

DOI: https://doi.org/10.7554/eLife.30005.001

*For correspondence:
santinj@missouri.edu

**Competing interests:** The authors declare that no competing interests exist.

## Introduction

For the brain to work properly, neurons must employ compensatory mechanisms to retain information in synapses and to maintain normal circuit function. These compensatory mechanisms are considered homeostatic if they regulate firing frequency and/or circuit activity around a set-point, but they also may oppose disturbances to neuronal function without being firmly homeostatic (*Turrigiano, 2012*). Homeostatic and compensatory plasticity maintains neuronal function through a suite of synaptic and intrinsic mechanisms (*Pozo and Goda, 2010*; *Schulz and Lane, 2017*; *Turrigiano, 2011*), with the ultimate goal being preservation of normal network function. The most prominent mechanisms used to counter disturbances in neuronal activity include synaptic scaling, a slow and global form of synaptic plasticity where excitatory synaptic strength increases or decreases equally across all synapses (*Turrigiano et al., 1998*), modulation of presynaptic neurotransmitter release (*Davis, 2013*), and altering voltage-gated membrane conductances (e.g. $Na^+$, $K^+$, and $Ca^{2+}$ channels) (*Desai et al., 1999*; *Ransdell et al., 2012*; *Wilhelm et al., 2009*).

Manipulations that perturb neuronal activity in vitro are, by far, the most common experimental approaches to evoke homeostatic plasticity (*Williams et al., 2013*), leaving many open questions as to what normal scenarios require homeostatic mechanisms to regulate circuit output and behavior. Only few physiological examples exist. In developing rodents, neurons in the visual cortex undergo synaptic scaling at the onset of visual experience (*Desai et al., 2002*). Also within the visual system, neurons in the retinotectal circuit of the aquatic frog, *Xenopus laveis*, homeostatically decrease their intrinsic excitability to compensate for developmental increases in excitatory synaptic inputs

**eLife digest** Neurons in the brain communicate using chemical signals that they send and receive across junctions called synapses. To maintain normal behavior over time, circuits of neurons must reliably process these signals. A variety of nervous system disorders may result if they are unable to do so, as may occur when neural activity changes as a result of disease or injury.

The processes underlying the stability of a neuron's synapses is referred to as "homeostatic" synaptic plasticity because the changes made by the neuron directly oppose the altered level of activity. In one form of homeostatic plasticity, known as synaptic scaling, neurons modify the strength of all of their synapses in response to changes in neural activity.

There is substantial experimental evidence to show that in young animals, neurons that communicate using a chemical called glutamate undergo synaptic scaling in response to artificial changes in activity. It had not been directly shown that synaptic scaling protects the neural activity of adult animals in their natural environments, in part, because neural activity in most healthy animals generally only goes through small changes. However, the neurons in the brain that cause the breathing muscles of bullfrogs to contract are ideal for studying homeostatic plasticity because they are naturally inactive for several months when frogs hibernate in ponds during the winter. During this time, the bullfrogs do not need to use their lungs to breathe because enough oxygen passes through their skin to keep them alive.

Santin et al. have now observed synaptic scaling of glutamate synapses in individual bullfrog neurons that had been inactive for two months. Further experiments that examined the activity of the breathing control circuit in the brainstem provided evidence that synaptic scaling leads to sufficient amounts of neural activity that would activate the breathing muscles when frogs emerge from hibernation. Therefore neural activity after prolonged, natural inactivity relies on synaptic scaling to preserve life-sustaining behavior in frogs.

These results open up new questions: mainly, how do synaptic scaling and other forms of homeostatic plasticity operate in animals as they experience normal variations in neural activity? Determining how homeostatic plasticity works normally in an animal will help us to understand what happens when plasticity mechanisms go wrong, as is thought to occur in several human nervous system diseases including nervous system injury, Alzheimer's disease, and epilepsy.

DOI: https://doi.org/10.7554/eLife.30005.002

(*Pratt and Aizenman, 2007*). Additionally, plasticity in the dynamic regulation of synaptic strength seems to compensate for variability in neuron number in the crustacean stomatogastric ganglion (*Daur et al., 2012*). However, rather than uncovering responses to normal, expected physiological challenges, most investigations of homeostatic plasticity dramatically perturb neuronal activity with artificial (e.g. pharmacological and genetic manipulations) or pathological (e.g. sensory denervation/stimulation, injury) modalities. While these synthetic challenges evoke striking compensatory or homeostatic responses in neurons in vivo (*Aizenman et al., 2003*; *Braegelmann et al., 2017*; *Echegoyen et al., 2007*; *Frank, 2014*; *Gonzalez-Islas and Wenner, 2006*; *Hengen et al., 2013*; *Kline et al., 2007*; *Knogler et al., 2010*; *Lambo and Turrigiano, 2013*; *Rajman et al., 2017*), how stabilizing mechanisms activated by these kinds of disturbances relate to non-pathological physiological scenarios is unclear, especially in mature animals. Animals with life histories involving profound disturbances to neuronal activity may provide new insights into the physiological necessity of such mechanisms underlying stability of neurons in functional circuits.

Here, we take advantage of an adult animal (American bullfrogs, *Lithobates catesbeianus*) that normally undergoes drastic and prolonged reductions in neuronal activity to understand if compensatory mechanisms commonly evoked during artificial deprivation of neuronal firing occur in an environmentally relevant setting. We demonstrate that a well-described mechanism of homeostatic plasticity, up-scaling of excitatory synapses, occurs in motoneurons innervating a primary respiratory muscle from bullfrogs after 2 months in a submerged-aquatic, overwintering habitat: a natural environment that induces complete respiratory motor inactivity (*Santin and Hartzler, 2017*). Strikingly, we further identify that increased excitatory synaptic strength onto these motoneurons enhances population motoneuron output from the respiratory network, thereby acting to preserve respiratory

motor drive to critical respiratory muscles under conditions when breathing would be obligatory at warm temperatures after 2 months without breathing movements (*Santin and Hartzler, 2016a*). As failure to stabilize motor output from the brainstem to respiratory muscles has fatal consequences, these results implicate up-scaling of excitatory synapses on respiratory motoneurons as a critical mechanism for an adult vertebrate experiencing prolonged bouts of neural inactivity in a natural context.

## Results

American bullfrogs often overwinter in ice-covered ponds with no need to use their lungs for breathing during cold winter months due to adequate skin gas exchange at low temperatures (*Tattersall and Ultsch, 2008*). Although frogs retain a relatively high locomotor capacity during overwintering submergence (*Tattersall and Boutilier, 1999*), important muscles of the respiratory apparatus, buccal floor constrictors and glottal dilators, are inactive during cold-submergence (*Santin and Hartzler, 2017*). Despite months of respiratory motor inactivity, upon forced emergence at warm temperature bullfrogs immediately exhibit breathing movements, ventilate to match resting metabolic demands, increase ventilation during exposure to low oxygen, and generate respiratory motor output from the brainstem (*Santin and Hartzler, 2016a*; *Santin and Hartzler, 2016b*). Largely normal ventilatory behaviors after months of motor inactivity led us to hypothesize that preservation of respiratory motoneuron output from the brainstem to breathing muscles may rely on slow acting, global mechanisms that compensate for neuronal inactivity (i.e. homeostatic plasticity). To test this hypothesis, we backfilled vagal motoneurons that innervate a primary respiratory muscle, the glottal dilator involved in regulating airflow into and out of the lung in frogs (*Gans et al., 1969*), with fluorescent dye (*Figure 1*). We then measured synaptic and intrinsic neuronal properties from fluorescently labeled respiratory motoneurons in brainstem slices at 23°C (a temperature requiring lung breathing) in control bullfrogs and those that experienced ∼2 months of respiratory motor inactivity during overwintering-like conditions. This allowed us to determine the extent to which winter inactivity leads to expression of compensatory mechanisms in respiratory motoneurons.

Increases in excitatory synaptic transmission occur during inactivity/AMPA-glutamate receptor blockade in vitro (*Fong et al., 2015*; *Turrigiano et al., 1998*) and in vivo (*Hengen et al., 2013*; *Knogler et al., 2010*). Therefore, we first sought to determine whether winter inactivity results in a compensatory increase in excitatory synaptic transmission by assessing the quantal amplitude (i.e. the postsynaptic response to spontaneous release of neurotransmitter single vesicles recorded in voltage clamp, termed miniature excitatory post synaptic currents [mEPSCs]). Consistent with this hypothesis, the amplitude and charge transfer of mEPSCs mediated by AMPA-glutamate receptors were increased in respiratory motoneurons after overwintering inactivity (*Figure 2A–D*). In contrast, we did not observe changes in mEPSC frequency (*Figure 2E*), suggesting a similar number of glutamate release sites onto respiratory motoneurons in brainstem slices. Although the mEPSC amplitude is prone to dendritic filtering, charge transfer (i.e. mEPSC area) is more robust against space clamp errors (*Spruston et al., 1993*). Thus, increases in both amplitude and charge transfer of the mEPSC suggest an enhancement of post-synaptic AMPA receptor function rather than changes in dendritic filtering properties of motoneurons (*Han and Stevens, 2009*). Consistent with this assertion, parameters influencing and influenced by dendritic filtering, neuronal input resistance and mEPSC rise time, respectively, did not differ between control and winter inactivity motoneurons (*Figure 2F–G*). These results indicate that respiratory motoneurons have enhanced excitatory synaptic strength after winter inactivity presumably by up-regulating postsynaptic function of AMPA-glutamate receptors.

Increased excitatory synaptic strength of neurons responding to reductions in activity typically follows a multiplicative scaling relationship in dissociated culture (*Fong et al., 2015*; *Turrigiano et al., 1998*) and in vivo (*Knogler et al., 2010*; *Lambo and Turrigiano, 2013*), implying that all measurable synapses increase in strength relatively by the same extent. Thus, synaptic scaling maintains relative synaptic strength in the face of disturbances in neuronal activity. This is observed experimentally as a right-shift in the cumulative distribution of mEPSC amplitudes induced by inactivity that can be down-scaled mathematically by dividing the entire distribution of mEPSCs by a common scaling factor. The scaling factor is derived from the slope of a linear rank order plot of mEPSC amplitudes from control and activity-deprived distributions. In contrast to synaptic scaling, a rank order plot following long-term potentiation (LTP) - a rapid and synapse-specific form of synaptic plasticity- is

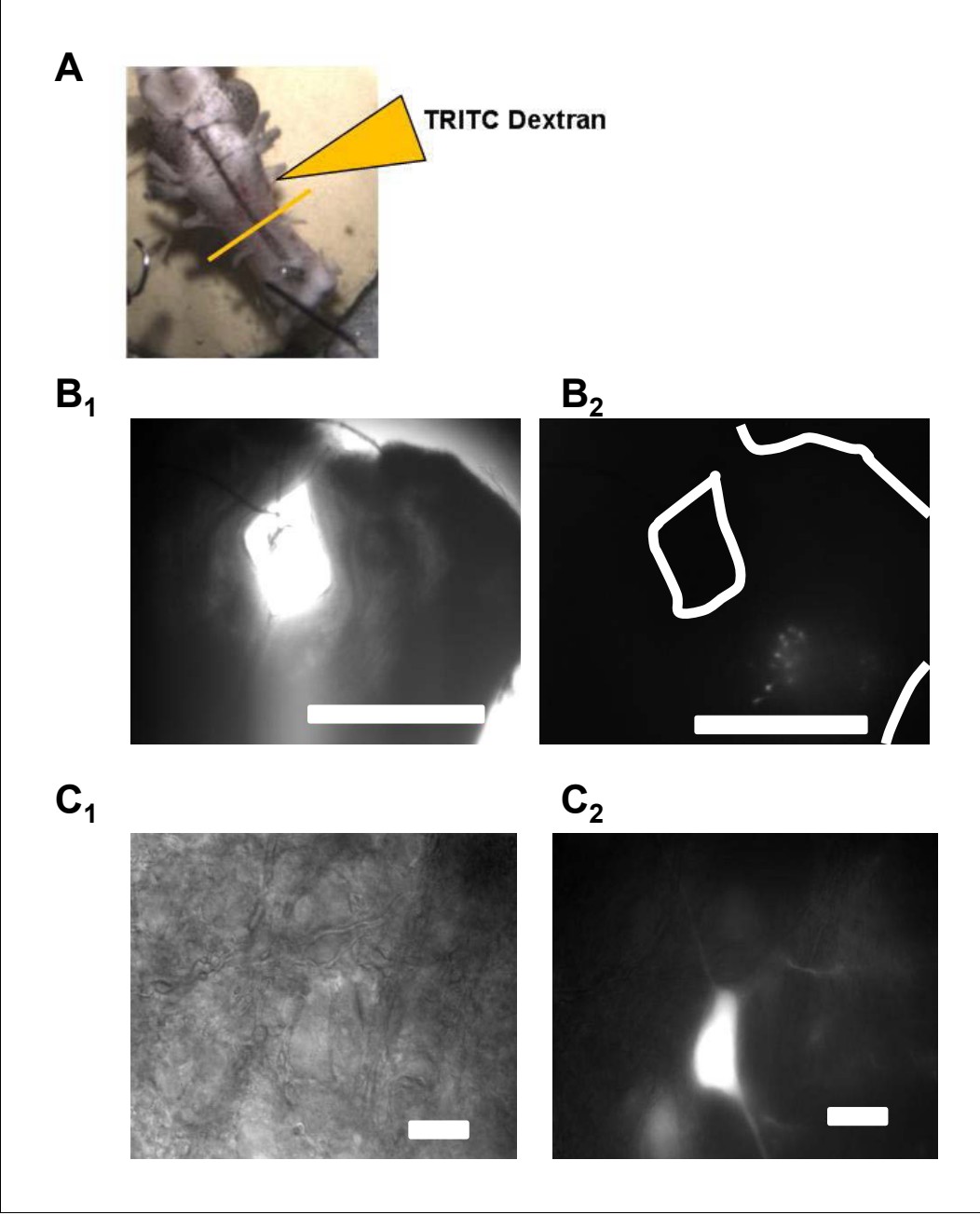

**Figure 1.** Vagal motoneurons predominately innervating the glottal dilator were labeled by backfilling the 4th (most caudal) root of the IX-X cranial nerve complex. (**A**) illustrates the tetramethylrhodamine-dextran (TRITC) backfill procedure. Orange line across brainstem approximates where brainstem slices were taken for experiments. (**B and C**) show examples fluorescent images of backfilled motoneurons loaded with TRITC-dextran at 4X and 60X. Bar markers are 1 mm (**B**) and 20 μm (**C**).

DOI: https://doi.org/10.7554/eLife.30005.003

better approximated by an exponential curve (*Gainey et al., 2009*). Do increases in the mEPSC amplitude scale multiplicatively following winter inactivity, an ecologically relevant, long-term perturbation of neuronal activity?

When rank ordering an equal number of mEPSC amplitudes from control and winter inactivity vagal respiratory motoneurons (black dots; 50 mEPSCs from 16 neurons in each group), the plot is well-fit by a linear regression ($r^2 = 0.99$; *Figure 3A*; dashed blue line), rather than an exponential

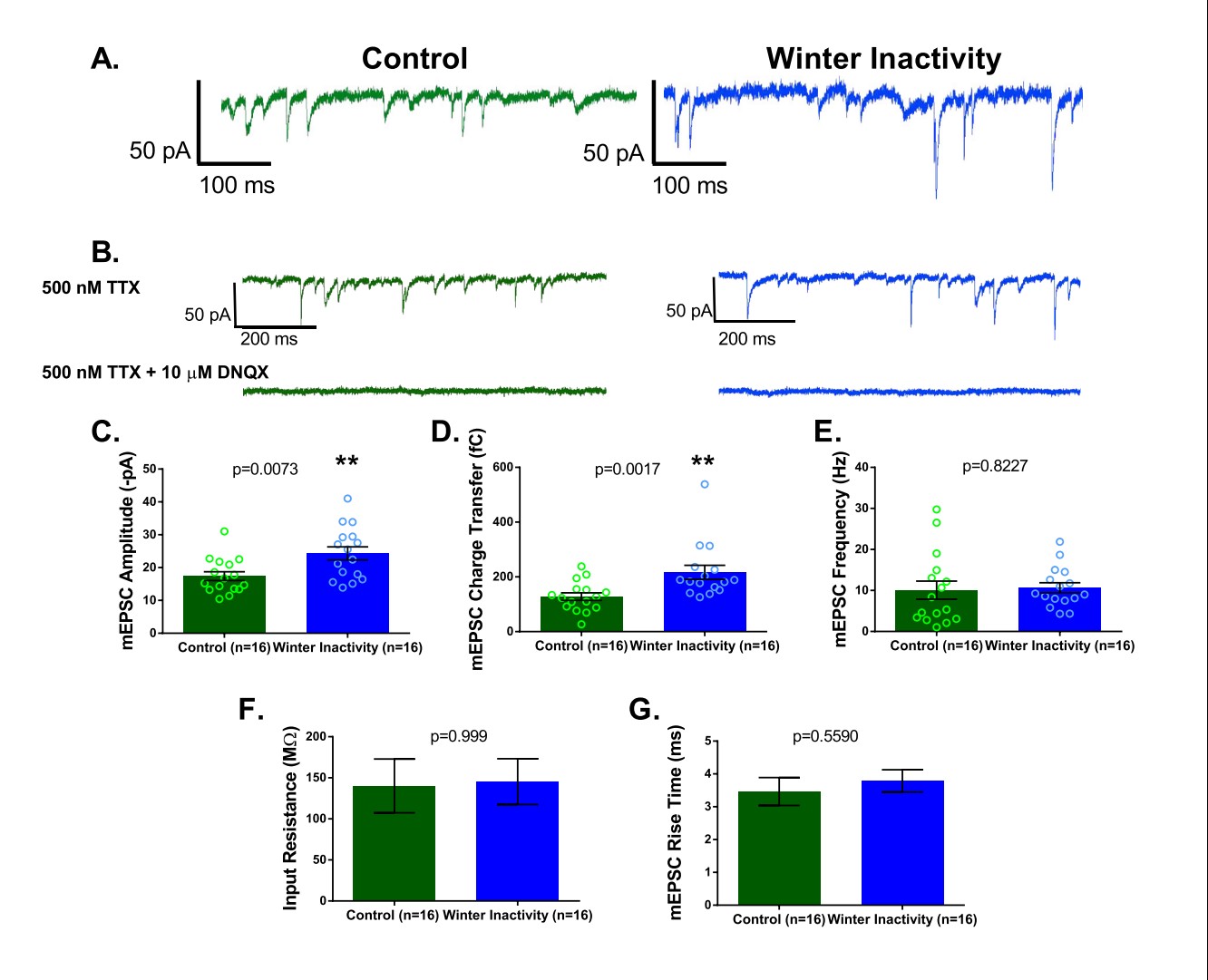

**Figure 2.** Respiratory motoneurons have increased synaptic strength after winter inactivity. 1A shows example voltage clamp traces of miniatures excitatory postsynaptic currents (mEPSCs; holding potential= −80 mV; 500 nM TTX) in control (green) and winter inactivity (blue) frogs. 2B shows that mEPSCs recorded at −80 mV are mediated by AMPA-glutamate receptors. The top panel of 1B shows example recordings one control and one winter inactivity motoneuron (different neurons than 2A) before (top panel) and after application of 10 µM DNQX, an AMPA-glutamate receptor antagonist (bottom panel). All mEPSCs that we could detect were blocked by DNQX. 2 C-E presents mean data for mEPSC amplitude (C; two-tailed unpaired t test), charge transfer (D; two-tailed unpaired t test with Welch's correction), and frequency (E; Two-tailed Mann Whitney test). mEPSC values for each neuron were obtained by averaging 1 min of data. Winter inactivity resulted in elevated mEPSC amplitude and charge transfer, but not frequency. Scattered circles in each bar graph are individual points that generate the mean. 2F-G shows mean data for input resistance (F; Two-tailed Mann Whitney test) and mEPSC rise time (G; Two-tailed unpaired t test). 2C-G analyzed n = 16 neurons for control (represented in green) and winter inactivity (represented in blue). N = 9 control frogs and N = 7 winter inactivity bullfrogs. Error bars are standard error of the mean (SEM). **p<0.01.
DOI: https://doi.org/10.7554/eLife.30005.004

curve ($r^2$ = 0.73; *Figure 3A*; dashed black line). If the slope of this line is greater than 1 (i.e., the unity line; *Figure 3A*; solid gray line), it suggests multiplicative scaling has occurred. We detected a slope of 1.51. The slope of the line from the regression is equal to the scaling factor, implying that all measurable excitatory synapses increased by a factor of 1.51 during winter inactivity. Indeed, when the right-shifted cumulative distribution of mEPSC amplitudes from winter inactivity motoneurons (*Figure 3B*; solid blue line; Kolmogorov-Smirnov test control *vs.* winter inactivity; p=2.2×10$^{-16}$) is divided by the scaling factor, the 'down-scaled' winter inactivity distribution (*Figure 3B*; dashed blue line) overlaps the control mEPSC amplitude distribution (*Figure 3B*; solid green line; Kolmogorov-

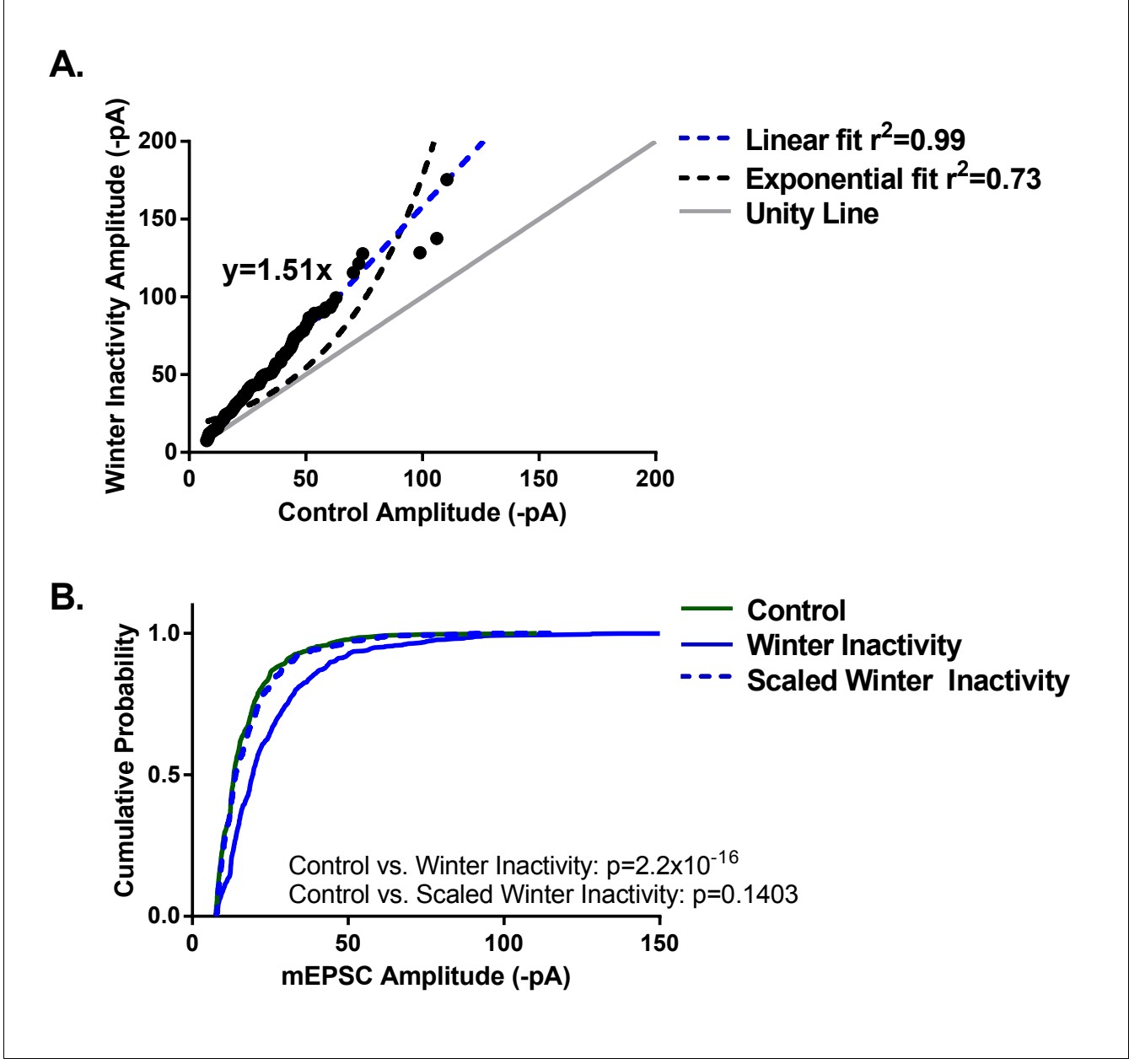

**Figure 3.** Winter inactivity scales up mEPSCs in respiratory motoneurons. *Figure 3A* shows linear, but not exponential, scaling of mEPSC amplitudes. Control and winter inactivity mEPSC amplitudes were rank ordered (50 mEPSCs in n = 16 neurons from each group; 800 total points) and fit with a linear regression (dashed blue line; $r^2$ = 0.99). The rank ordered plot was not well-fit by an exponential curve (dashed black line; $r^2$ = 0.73). The gray unity line with a slope of 1 represents the relationship if winter inactivity did not result in increased mEPSC amplitude; however, the linear regression fit of ranked ordered mEPSCs produced a slope of 1.51. *Figure 3B* shows cumulative probably histograms of mEPSC amplitudes from control neurons (solid green line), winter inactivity neurons (solid blue line), and the winter inactivity distribution down-scaled by the scaling factor, 1.51 (dashed blue line). Because down-scaled mEPSCs below the detection threshold (7.5 pA in our analysis) cannot overlap with the control distribution (*Kim et al., 2012*), scaled mEPSCs below 7.5 pA were not included in analysis. Kolomogorov-Smirnov tests revealed that control and winter inactivity are significantly different ($p < 2.2 \times 10^{-16}$), but control and scaled winter-inactivity distributions were not significantly different (p=0.1403). This provides evidence that multiplicative synaptic scaling occurred across the distribution of measurable mEPSCs.
DOI: https://doi.org/10.7554/eLife.30005.005

Smirnov test control vs. scale winter inactivity; p=0.1403). Therefore, excitatory synapses on inactive respiratory motoneurons 'scale up' during winter inactivity.

In addition to synaptic scaling, reduced activity may influence the function of intrinsic ion channels, including $Na^+$, $K^+$, $Ca^{2+}$, and non-selective cation channels, to oppose reductions in neuronal and network activity. Compensatory changes in the intrinsic membrane currents carried by ion channels during inactivity typically manifest as enhanced excitability in response to injected current (*Desai et al., 1999*; *Echegoyen et al., 2007*; *Lambo and Turrigiano, 2013*; *Wilhelm et al., 2009*). Therefore, we evaluated action potential firing in respiratory motoneurons to determine whether intrinsic changes, in addition to synaptic scaling, may be involved in the compensatory response to winter inactivity. To assess intrinsic excitability, we measured firing frequency of respiratory motoneurons in response to step increases in injected current. *Figure 4A* shows example recordings of responses to current injections from control (green traces; top) and winter inactivity (blue traces; bottom) motoneurons. We did not observe differences in firing frequency at any amount of current (*Figure 4B*). The frequency-current (F-I) gain did not differ between both groups of motoneurons (*Figure 4B–C*). Additionally, resting membrane potential did not differ between control and winter inactivity motoneurons (control: $-57.62 \pm 1.74$ mV vs. winter inactivity: $-57.13 \pm 1.41$ mV; p=0.8269; $T_{30} = 0.2206$). These results suggest that these motoneurons do not enhance their excitability in response to winter inactivity.

Thus far, our results indicate that up-scaling of AMPA glutamate-receptors increases the synaptic strength of respiratory motoneurons in response to winter inactivity. How may this affect respiratory behavior immediately following overwintering? In bullfrogs and other vertebrate animals, central pattern generating (CPG) networks in the brainstem underlie the neural activity of respiratory movements (*Baghdadwala et al., 2015*; *Feldman et al., 2013*). This respiratory CPG activity is transmitted to cranial and spinal motoneurons, in part, via excitatory glutamatergic synapses (*Greer et al., 1991*; *Kottick et al., 2013*) to produce the motor output that activates respiratory muscles. We hypothesized that synaptic scaling on respiratory vagal motoneurons receiving respiratory CPG input could increase the reliance of AMPA-receptor transmission of the vagal respiratory motor outflow in bullfrogs after winter inactivity. To test this possibility, we extracellularly recorded respiratory-related vagal motoneuron population discharge from the cut nerve root (i.e. fictive breaths exiting the brainstem via the axons of vagal motoneurons) in rhythmically-active, in vitro brainstem-spinal cord preparations (*Figure 5A*); a preparation that produces respiratory motor output similar to that observed in intact bullfrogs (*Hedrick, 2005*). In rhythmic respiratory preparations, synaptic properties of motoneurons (and premotoneurons) can be determined by assessing relative changes in the amplitude of the integrated fictive breath after application of receptor antagonists, while the activity of the rhythm generating circuits can be assessed through the frequency of fictive breaths (for example *Greer et al., 1991*; *Johnson et al., 2002*). Using the same in vitro preparation from bullfrogs, *Chen and Hedrick (2008)* showed that a low concentration of AMPA-receptor antagonist does not influence the amplitude of fictive breaths, while the frequency is more sensitive to inhibition of AMPA-receptors. To test the hypothesis that winter inactivity enhances the reliance of AMPA-receptor transmission on vagal motoneurons and to explore the possibility of altered AMPA-receptors in rhythm generating circuits, we measured the amplitude and frequency of fictive-breaths via vagal motoneurons to assess differences in the sensitivity to an AMPA receptor antagonist, DNQX, between control and winter inactivity bullfrogs. To be consistent with our results demonstrating up-scaling of AMPA receptors at the level of the vagal motoneuron after winter inactivity (*Figures 2–3*), the amplitude fictive breath carried by vagal motoneurons should be more sensitive to inhibition of AMPA receptors.

First, we analyzed the amplitude of the integrated motoneuron population discharge of the vagus nerve (cranial nerve X; CN X) during lung-related respiratory bursts (*Figure 5A*) to infer whether or not scaling of AMPA-receptors on vagal motoneurons increased the reliance of AMPA-receptors for transmitting respiratory output. An inherent limitation to measuring the amplitude of the extracellular integrated motoneuron population discharge from the cut nerve root in vitro is that only relative comparisons can be made. This precludes us from comparing absolute fictive breath amplitudes at baseline between control and winter inactivity frogs. Therefore, we rely on relative changes in burst amplitude in response to a low concentration of DNQX that did not silence respiratory bursts (4 µM) to infer differences in the reliance on AMPA-receptors for producing 'normal,' baseline respiratory motoneuron population amplitude in each group of frogs. In control bullfrogs, AMPA-receptor

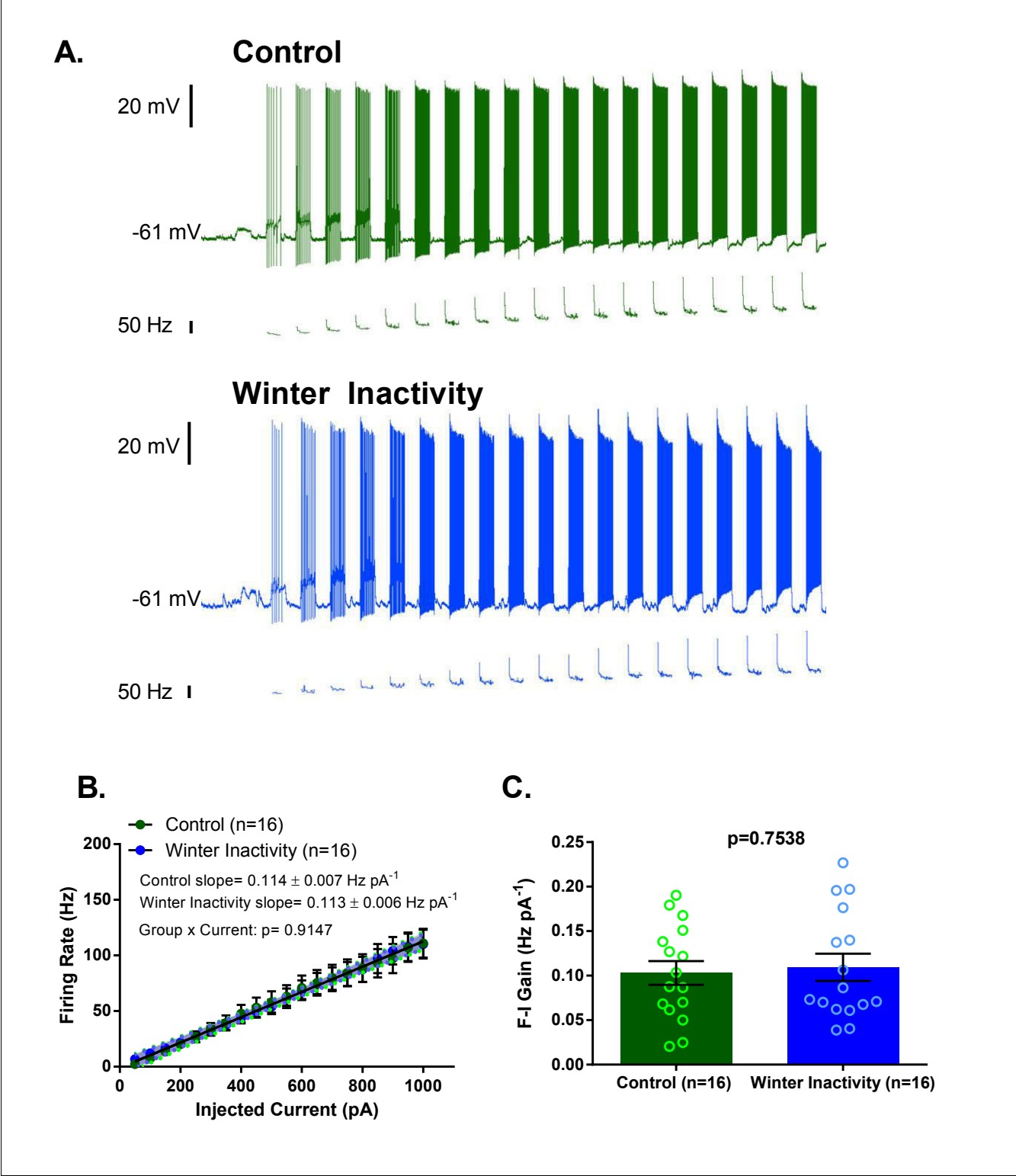

**Figure 4.** Increases in F-I (frequency-current) gain do not accompany up-scaling of excitatory synapses. *Figure 4A* shows example current clamp recordings of control (green) and winter inactivity (blue) motoneurons during step current injections (+50 pA X 20 steps; 0.5 s per step). *Figure 4B* shows mean firing rates during 0.5 s, step increases in current in control and winter inactivity motoneurons (n = 16 neurons for each group). There was no difference in in the slope of mean firing frequency-current relationship (p=0.9147; Analysis of Covariance). The shaded area around the linear regression lines are 95% confidence bands. *Figure 4C* shows that mean firing frequency-current relationships (F-I gain) from individual neurons do not different between groups (two-tailed unpaired t test). *Figure 4B–C* analyzed n = 16 neurons for control (represented in green) and winter inactivity

*Figure 4 continued on next page*

Figure 4 continued

(represented in blue). N = 9 control frogs and N = 7 winter inactivity bullfrogs. Scattered circles in each bar graph are individual points that generate the mean. Error bars are standard error of the mean (SEM).

DOI: https://doi.org/10.7554/eLife.30005.006

inhibition did not change the amplitude and area of the respiratory-related CN X population discharge (*Figure 5B,C–D*; green traces and bars), corroborating previous results in bullfrogs (*Chen and Hedrick, 2008*). This was not the case for bullfrogs following winter inactivity. In stark contrast, CN X population output underwent a reduction in amplitude and area during exposure to AMPA-receptor blockade (*Figure 5B,C–D*, blue traces and bars) and tended to recover toward baseline values following ~1 hr of DNQX washout (*Figure 5—figure supplement 1*). These results indicate that full transmission of respiratory CPG activity to vagal motoneurons requires an increase in synaptic strength following winter inactivity. With respect to fictive breathing frequency, brainstem-spinal cord preparations from winter inactivity bullfrogs produced an elevated number of respiratory bursts at baseline compared to control bullfrogs (*Figure 5B & E₂*). During exposure of the brainstem-spinal cord preparations to a low concentration of DNQX, both control and winter inactivity bullfrogs decreased fictive-breathing frequency by ~80% (*Figure 5 E1*), suggesting AMPA receptor-sensitive components of the respiratory rhythm generator did not differ. Collectively, our findings imply that up-scaling of AMPA-glutamate receptors on vagal motoneurons preserves respiratory motor output because, in its absence (i.e., AMPA-receptor inhibition), the amplitude of vagal motoneuron population discharge was approximately 40% less than baseline on average, while respiratory motor amplitude was unaffected by the same concentration of AMPA receptor blocker in control bullfrogs.

## Discussion

The mechanisms underlying how stable neuronal function emanates from inherently unstable components have been of intense interest to neuroscientists for over two decades (*Abbott and LeMasson, 1993*; *Siegel et al., 1994*; *Turrigiano et al., 1994*). There has been an explosion of mechanisms - postsynaptic, presynaptic, cell-autonomous, non-neuronal, etc - generated by in vitro, in vivo, and in silico approaches that paint an immensely complex picture of the genesis and maintenance of stable neuronal function (*O'Leary et al., 2014*; *Schulz and Lane, 2017*; *Stellwagen and Malenka, 2006*; *Turrigiano, 2012*). A sobering reality is that only a few studies experimentally relate mechanisms of homeostatic plasticity to the stabilization of behaviorally relevant circuit function [e.g. (*Gonzalez-Islas and Wenner, 2006*; *Knogler et al., 2010*; *Lane et al., 2016*). Of such studies, most opt for tractability at the expense of reality by using drastic experimental perturbations that most organisms will never experience. Although these mechanistic studies are critical, it is worth considering how animals experiencing analogous situations utilize those mechanisms for survival in their environments. After all, it is the animal in its natural environmental which natural selection acts upon.

Because respiratory motor output of bullfrogs is silent under submerged-overwinter conditions (*Santin and Hartzler, 2017*), they provided an ideal platform to test whether mechanisms that stabilize neuronal output over chronic time scales respond to prolonged inactivity experienced normally by an animal. In a sense, this serves as a natural analogue for experiments that dramatically reduce neural activity via pharmacological tools or injury because bullfrogs do not use neural mechanisms of ventilatory control for potentially several months during overwintering submergence. We found that respiratory vagal motoneurons underwent up-scaling of excitatory synapses in response to winter inactivity. This result is intriguing because up-scaling commonly occurs in response to experimental decreases in neuronal activity and/or synaptic transmission (*Fong et al., 2015*; *Lambo and Turrigiano, 2013*; *Turrigiano et al., 1998*), indicating that up-scaling of AMPA-receptors in response to inactivity-related stimuli is a physiologically relevant response in an animal normally experiencing extremely low levels of neuronal activity. Strikingly, we also observed that vagal motoneuron population output driven by respiratory CPG activity was ~40% less than baseline when we, presumably, blocked the effects of scaling with a sub-saturating concentration of AMPA-receptor antagonist after winter inactivity. This contrasts with no effect of AMPA-receptor block on respiratory motor amplitude in control bullfrogs (*Figure 5*). Given that amplitude of population motoneuron firing

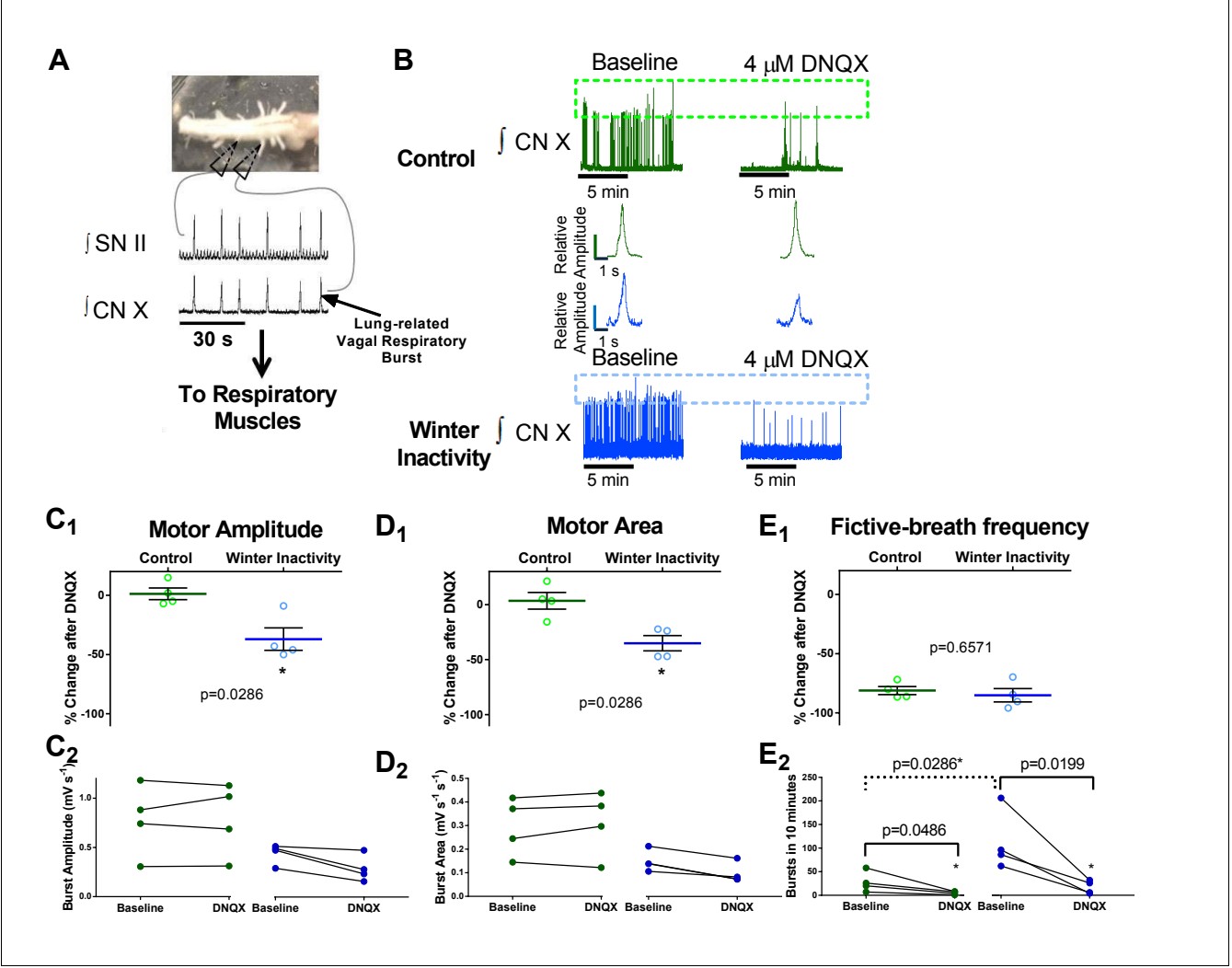

**Figure 5.** Increases in AMPA-glutamate receptors sustain motor output from the respiratory network after winter inactivity. *Figure 5A* illustrates the bullfrog in vitro brainstem-spinal cord preparation. In bullfrogs in vivo, lung ventilation cycles involve activation of both glottal dilator (value to the lung) and buccal floor pump muscles. The general motor behavior associated with breathing persists in the isolated brainstem-spinal cord in vitro and can be recorded with extracellular suction electrodes through cut spinal nerve II (hypoglossal nerve; top) that innervates the buccal floor, and the caudal portion of cranial nerve X (vagus; bottom) that innervates the glottis (*Sakakibara, 1984a*). Since activation of both groups of motoneurons are required for lung ventilation, coactivation of buccal floor and glottal-related motor outflow confirms respiratory-related central pattern generator activity associated with lung ventilation (i.e. a fictive lung breath; *Figure 5A*) (*Sanders and Milsom, 2001*). Here, we analyzed respiratory bursts recorded from the CN X nerve root as it contains the glottal dilator motoneurons (*Kogo et al., 1994*) that we showed to exhibit up-scaling of AMPA receptors. *Figure 5B* shows example CN X root recordings from control (green; top) and winter inactivity (blue; bottom) in vitro brainstem-spinal cord preparations before (left) and after application of 4 μM DNQX (right). Dashed boxes across the compressed recordings represent minimum and maximum burst amplitudes with each preparation. Both control and winter inactivity preparations underwent relatively similar decreases in respiratory burst frequency during bath application of DNQX; however, only preparations from winter inactivity bullfrogs experienced a decreased in burst amplitude. This can be further observed in the expanded example integrated motoneuron population burst in the inset. *Figure 5C1–E1* shows mean changes in fictive lung burst amplitude, area, and frequency in response to 4 μM DNQX expressed as a percent of baseline (n = 4 brainstems per group; two-tailed Mann Whitney test). In winter inactivity bullfrogs, DNQX led to a significant decrease in burst amplitude and area compared to control bullfrogs. *Figure 5C2–E2* show raw values used to generate each of the means in *Figure 5C1–E1*. DNQX caused a decrease in fictive lung burst frequency in both groups of frogs despite winter inactivity frogs starting from a higher baseline (*Figure 5E2*). Absolute comparisons for burst amplitude and area characteristics were not performed as these are only relative measures. Means for each analysis are shown as a solid green or blue line. Scattered circles in each graph are individual points that generate the mean. Error bars are standard error of the mean (SEM). *p<0.05.

DOI: https://doi.org/10.7554/eLife.30005.007

The following figure supplement is available for figure 5:

*Figure 5 continued*

**Figure supplement 1.** Respiratory burst amplitude from each winter inactivity bullfrog tends to recover upon washout (A. winter inactivity group: paired t test; p=0.0351; DNQX vs. washout) and returns to near-control values (one sample t test; p=0.0333 compared to 100% during DNQX and p=0.0609 during washout, n = 4).

DOI: https://doi.org/10.7554/eLife.30005.008

corresponds with drive to respiratory muscles in vivo (*Sakakibara, 1984b*), up-scaling seems to ensure adequate motor output to critical breathing muscles. Since lung ventilation satisfies gas exchange requirements immediately after winter inactivity in intact, unrestrained bullfrogs (*Santin and Hartzler, 2016a*), our findings suggest that scaling of AMPA receptors on respiratory motoneurons may contribute to normal ventilation after months without lung breathing.

Several lines of evidence suggest this is the case. First, we observed an increase in mEPSC amplitude and charge transfer without changes in neuronal input resistance and mEPSC rise time, indicating that increased function of AMPA-glutamate receptors, rather than differences in electrotonic properties, explain the increase in mEPSC amplitude (*Figure 2*) (*Han and Stevens, 2009*). Whether post-synaptic enhancement of AMPA-receptor function (via elevated expression or post-translational modification) or a presynaptic synaptic mechanism (*Liu et al., 1999*) causes the larger mEPSC remains an open question. Second, ranked ordered mEPSCs from control and winter inactivity distributions were well-fit by a linear regression, a hallmark of synaptic scaling, instead of an exponential curve, which describes the rank order relationship following LTP (*Figure 3A*) (*Gainey et al., 2009*). Third, dividing the winter inactivity mEPSC amplitude distribution by the scaling factor obtained from the linear fit of ranked order mEPSC amplitudes produced a 'down-scaled' distribution that overlays the control mEPSC amplitude distribution (*Figure 3B*) (*Turrigiano et al., 1998*). Points two and three imply that AMPA-receptors scaled slowly and globally throughout the duration of winter inactivity, rather than rapidly through LTP-like mechanisms during the time between removal of the frog from the simulated winter environment and ~3 hr later when the recordings were made. However, the timing of induction and completion of up-scaling in vagal respiratory motoneurons during the course of the 2-month period without breathing is unresolved. Finally, increased synaptic strength enhanced the contribution of AMPA-receptors for producing respiratory motoneuron discharge. Unlike control bullfrogs, vagal motoneurons receiving respiratory CPG input reduced motoneuron population amplitude by ~40% after winter inactivity upon exposure to a low concentration of AMPA-receptor antagonist (*Figure 5*). Consistent with a compensatory or homeostatic response, this result suggests that respiratory output leaving the brainstem through the vagus nerve would have been absolutely smaller in bullfrogs after winter inactivity if scaling did not occur. Since motoneuron population discharge increases proportionally with outflow to respiratory muscles in vivo (*Sakakibara, 1984b*), we collectively interpret our results to mean that scaling of AMPA-receptors on respiratory motoneurons is part of a strategy used to preserve respiratory motor output that produces appropriate lung breathing after winter inactivity.

Although the amplitude of vagal motoneuron output became increasingly reliant on AMPA-receptors following winter inactivity, upregulation of AMPA-receptors probably does not underlie the facilitation of fictive-breathing frequency that we observed (*Figure 5E* present study, *Santin and Hartzler, 2016b*). This implies that respiratory rhythm generating circuits use distinct mechanisms of compensation to enhance the frequency of respiratory-related discharge following winter inactivity. Although the mechanisms are presently unknown, several, potentially interacting, processes could underlie enhanced burst frequency in vitro after winter inactivity. These mechanisms may include, but are not limited to (1) decreasing the inhibitory GABA/glycinergic tone within rhythm generating circuits (*Straus et al., 2000*), (2) altering the neuromodulatory state of rhythm generating circuits, and (3) decreasing the $O_2$ sensitivity of the respiratory control system that may act to tonically depress fictive lung bursts (*Santin and Hartzler, 2016b*; *Winmill et al., 2005*). Although the mechanisms leading to enhanced fictive lung burst frequency remain untested, it is apparent that a rich repertoire of compensatory mechanisms may ensure reliable action of the different components of the respiratory control system after months of inactivity.

Exciting questions remain for homeostatic plasticity that can only be understood using animals in natural environments; mainly, how do such compensatory mechanisms operate in natural contexts? An inability to understand principles linking induction and maintenance of these compensatory

mechanisms with normal physiological challenges makes it difficult to imagine that these mechanisms can be extrapolated and understood in the context of human pathologies for which they may play a role (*Beck and Yaari, 2008*; *D'Amico et al., 2014*; *Rajman et al., 2017*). Recently, factors associated with physiological state (e.g. sleep-wake) have been suggested to induce synaptic scaling (*Diering et al., 2017*) implying that neuronal circuits in intact animals use homeostatic and compensatory mechanisms that extend beyond the simple paradigms in which they are commonly studied. The respiratory motor network of bullfrogs following winter inactivity provides a tractable and ecologically relevant model to address major questions regarding preservation of nervous system output. For example, how do neurons 'know' when to scale their synapses during physiological challenges (or even when to stop scaling), do specific or multiple interacting stimuli trigger synaptic scaling during natural perturbations in vivo, and why do different components of a behavioral control system apparently use distinct compensatory mechanisms? Only considering mechanisms in the context of realistic challenges encountered by the animals in which they exist can the plethora of mechanisms underlying brain stability be brought to life.

## Materials and methods

### Ethical approval

Experiments were approved by the Wright State University Institutional Animal Care and Use Committee (protocol number 1047).

### Experimental animal groups

Two groups of adult, female bullfrogs, *Lithobates catesbeianus*, were used in this study: (1) control frogs maintained at 22°C (n = 13) and (2) frogs experiencing an overwintering-like environment (n = 11). Control bullfrogs were maintained in a plastic tank containing 22°C aerated water, provided crickets two times per week, could access wet and dry areas, and were kept on a 12 hr:12 hr light: dark cycle. Control frogs were maintained in this environment for at least 1 week before experiments. Winter inactivity frogs were kept in a plastic tank under the same conditions as control frogs for ~1 week before water temperature was gradually cooled to 2°C over 6 weeks (~3.3°C decrease per week) in a walk-in, temperature-controlled environmental chamber. Frogs were fed twice per week during the cooling phase until water temperature reached ~7°C, and then food was withheld because frogs no longer ate. The light:dark cycle was gradually shifted from 12 hr:12 hr to 10 hr:14 hr over the 6-week cooling phase to simulate day length changes that occur during the winter. Once water temperature reached 2°C, air access was denied using a plastic screen placed in the tank. After 8 weeks of submergence, experiments commenced.

### Preparation of brainstem slices and labeling of respiratory motoneurons

To generate brainstem slices containing vagal motoneurons, the brainstem-spinal cord was dissected as previously described (*Santin and Hartzler, 2016b*). The dissection was performed in bullfrog artificial cerebrospinal fluid (aCSF; concentrations in mM: 104 NaCl, 4 KCl, 1.4 MgCl$_2$, 7.5 glucose, 40 NaHCO$_3$, 2.5 CaCl$_2$ and 1 NaH$_2$PO$_4$, and gassed with 90% O$_2$, 1.3%CO$_2$, balance N$_2$; pH = 7.8; CO$_2$/pH values reflect normal for bullfrogs). The glottal dilator muscle gates airflow into and out of the lungs during ventilation in anuran amphibians (*Gans et al., 1969*) and receives its innervation from the laryngeal branch of the vagus nerve that arises the fourth rootlet of the IX-X complex in anurans (*Stuesse et al., 1984*; *Yamaguchi et al., 2003*; *Zornik and Kelley, 2007*). The fourth root of the IX-X nerve is composed of axons mostly projecting to laryngeal muscles (*Simpson et al., 1986*). To label motoneurons involved in lung ventilation, following dissection of the brainstem-spinal cord, the 4th branch of the IX-X cranial nerve root was isolated from the rest of the root using fine, spring scissors and then cutting the first three branches close to their exit point from the brain. A similar preparation has been used in *Xenopus laevis* to prepare slices containing laryngeal motoneurons (*Yamaguchi et al., 2003*). The brainstem-spinal cord was then pinned to a 6 mL Sylard (Dow Corning, Midland, MI)-coated dish filled with aCSF. The 4$^{th}$ branch of the IX-X complex was drawn into a glass pipette using suction. aCSF was removed from the pipette using polyethylene 50 tubing connected to syringe and was replaced with 2–3 µL of 10% tetramethylrhodamine-dextran,

3000 MW (ThermoFisher Scientific, Waltham, MA). The dye was allowed to diffuse for 3 hr while the brain was superfused with gassed aCSF at 10 mL min$^{-1}$. We found that 3 hr was sufficient to label motoneurons (*Figure 1*).

Following the 3 hr incubation period, we glued the brainstem to an agar block and then cut 300-μM-thick brainstem slices using a Vibratome tissue slicer (Leica Microsystems, Buffalo Grove, IL) in cold, gassed aCSF as previously described (*Santin and Hartzler, 2016b*). Given the complex organization of the cranial motor nuclei in amphibians (*Matesz and Székely, 1978*), we selected slices with a high probability to contain laryngeal motoneurons that drive the glottal dilator based on the following anatomical rationale. The 4th root of the IX-X complex contains the axons innervating the glottal dilator in anurans, but this root also contains axons projecting motoneuron other peripheral targets. However, there is less overlap in the caudal brainstem near the obex (*Stuesse et al., 1984*). Furthermore, laryngeal motoneurons are morphologically distinct from gastric and cardiac vagal motoneurons (double the soma diameter of the long axis) that reside in the same area (*Matesz and Székely, 1996*). To maximize the likelihood that we recorded from laryngeal motoneurons that innervate the glottal dilator muscle, we (1) used brainstem slices within 600 μm rostral to the obex (*Stuesse et al., 1984*) and (2) selected labeled neurons with a long soma diameter >20 μm as these neurons are not visceral motoneurons (*Matesz and Székely, 1996*). Slices were then transferred to the 0.5 mL recording chamber and were superfused with gassed aCSF at 1–2 mL min$^{-1}$ during experiments. Preparation and neuron identification was the same for both groups of bullfrogs in this study.

## Whole-cell patch clamp recordings

Borosilicate glass pipettes were back-filled with intracellular solution containing (in mM): 110 potassium gluconate, 2 MgCl$_2$, 10 Hepes, 1 Na$_2$-ATP, 0.1 Na$_2$-GTP, 2.5 EGTA, pH 7.2 with KOH, and positioned over an AgCl$_2$-coated Ag wire with a resistance of 2–4 MΩ. The recording chamber was located under a fixed-stage microscope with a (Nikon, Elgin, IL) where the slice was visualized at 4x magnification with a Nikon Cool Snap (Nikon) to roughly identify the location of the X motor nucleus. Individual neurons were identified in this area using at 60X and labeling with tetramethylrhodamine-dextran was determined by fluorescence imaging (Lambda LS Xenon Lamp House with liquid light guide, Lambda 25 mm excitation filter wheel with SmartShutter, and Lambda 10–3 controller; Sutter Instrument Company, Novato, CA; C-FL G-E/C TRITC Filter Block DM 565, EX 540/25 (528-553), BA 620/60 (590-650). The recording electrode was placed near the neuron using a Burleigh micromanipulator (PCS 5000; Thorlabs, Newton, NJ) while applying positive pressure through the pipette. The pipette offset was zeroed before contacting the neuron. When the pipette touched the neuron, positive pressure was removed and slight negative pressure was applied by mouth until the formation of a gigaohm seal. Rapid negative pressure was applied by mouth to break the gigaohm seal and obtain whole-cell electrochemical access. Data were low-pass filtered at 2 kHz and acquired at 10 kHz for current clamp experiments and 100 kHz for voltage-clamp experiments with an Axopatch 200B amplifier, Digidata 1440A A/D converter, and Molecular Devices P10 Clampex software (Molecular Devices, Sunnydale, CA). Current and voltage-clamp recordings were analyzed offline using LabChart 8 (AD Instruments Inc., Colorado Springs, CO). All voltages from voltage- and current-clamp experiments were corrected for a liquid junction potential of −12 mV (pipette relative to the bath). Neurons with membrane voltages more negative than −45 mV and that contained >50 mV action potentials were used in analysis.

### Current and voltage-clamp protocols

Each neuron from both groups of bullfrogs (n = 16 neurons per group) was subjected to the same protocol. Once the neuron was determined to be viable in current clamp, a step protocol was applied to determine input resistance (R$_{in}$) and firing frequency-current relationship. The step protocol began with a 0.5 s, −150 pA step and then increased by 50 pA for a total of 24 steps (−150 pA to 1000 pA). After the step protocol, the neuron was voltage clamped at −80 mV and series resistance (R$_s$) was allowed to stabilize (control = 12.7 ± 4.7 MΩ *vs.* winter inactivity = 11.1 ± 4.2 MΩ;±1S. D.; p=0.3127; two-tailed, unpaired t test). R$_s$ was not compensated, but if R$_s$ changed by >20% during the recording or was large (>25 MΩ), the neuron was discarded from analysis. 500 nM tetrodotoxin (TTX) was bath applied for 5 min to block voltage-gated Na$^+$ channels, and then 1 min of

mEPSCs were recorded. Next, the neuron was superfused with 10 µM DNQX +500 nM TTX to confirm the AMPA-ergic identity of the mEPSCs. All mEPSCs from both groups of frogs recorded at −80 mV were blocked by 10 µM DNQX.

## Extracellular recording of respiratory-related nerve activity

The brainstem-spinal cord was dissected the same way as described for the preparation of the brainstem slices and pinned in a 6 mL, Sylgard-coated dish and superfused at 13 mL min$^{-1}$. Spinal nerve II (SN II; the hypoglossal nerve in anuran amphibians) and cranial nerve X (CN X; vagus) contain branches that innervate the respiratory muscles of amphibians; therefore, spontaneous, rhythmic activity recorded through these cranial nerves corresponds with respiratory rhythm/pattern generator central nervous system activity that drives breathing in intact frogs (*Sakakibara, 1984a*). SN II was pulled into a borosilicate glass suction electrode, and another glass electrode was attached to the caudal portion of CN X to maximize coverage of the 4th root that supplies the glottis. Glass electrodes were pulled using a two-stage micropipette puller (PC-10; Narishige, East Meadow, NY), broken to size to fit each rootlet, and then fire polished. Nerve activity was amplified ×1000 using differential amplifiers (DP-311; Warner Instruments, Hamden, CT), filtered (100–1000 Hz), full-wave rectified, integrated (time constant, 60 ms) and recorded using the Powerlab 8/35 data acquisition system (ADInstruments Inc., Colorado Springs, CO). Data shown here are the integrated traces.

### Extracellular recording protocol

The preparation was allowed 1 hr to stabilize following dissection and then recording of baseline respiratory bursting in gassed aCSF occurred for 20–30 min. After this baseline period, the preparation was exposed to 4 µM DNQX for 40 min. Upon washing with aCSF following recovery from DNQX, preparations recovered toward baseline amplitude in winter inactivity preparations (*Figure 5—figure supplement 1*).

## Data analysis

For voltage-clamp experiments, average amplitude (current measurement from baseline to peak), area (integral of the mESPC), rise time (time from baseline to peak), and frequency of mEPSCs (mEPSC per minute) (*Figure 2*) were analyzed from one minute of gap free recording following exposure to TTX using the peak analysis function on LabChart 8 (ADInstruments Inc., Colorado Springs, CO). We rejected events below 7.5 pA, as this value is about double the background noise and detection was unreliable. All events were inspected by eye to ensure the software detected mEPSCs. For rank ordering mEPSC amplitudes and construction of cumulative probability histogram (*Figure 3*), we used the first 50 mEPSCs from the minute of data obtained from each neuron. The amplitude average of the first 50 mEPSCs for each neuron was within ~1–2 pA of the average obtained using 1 mine of data. The winter inactivity distribution was downscaled by dividing all values in the distribution by the slope of the line of the rank order relationship (i.e. the scaling factor). We only included scaled values that were above the noise threshold of 7.5 pA, as mEPSC amplitudes below this value are not represented in the control distribution and would produce erroneous results in statistical analysis (*Kim et al., 2012*). For current-clamp experiments, input resistance was calculated as slope of the voltage-current relationship in response to negative current injections (−150 pA to 0 pA in 50 pA steps, 0.5 s steps). Membrane potential is reported as the voltage at the beginning of the step protocol. To establish the F-I gain, we averaged the firing frequency across the 0.5 s step from 0 pA to 1000 pA for each neuron. F-I gain for each neuron was also determined by taking the slope of the F-I relationship. For extracellular nerve recordings from the in vitro brainstem-spinal cord preparation, activity on the vagus nerve was classified as a respiratory burst if it occurred near-synchronously with the hypoglossal and if they were ~1 s in duration. We analyzed 10 min of burst data in control before application of 4 µM DNQX, the last 10 min of burst data after 40 min of exposure to 4 µM DNQX, and 10 min of data ~ 1 hr after washing DNQX. Burst properties were determined using LabChart 8 (ADInstruments Inc., Colorado Springs, CO, USA). Fictive-breath amplitude was analyzed as the maximum height subtracted from the baseline at the start of each burst of the integrated neurogram. Fictive-breath area was obtained from the area under the curve of the integrated nerve burst. Fictive-breath properties in DNQX and washout are expressed as a percent change from baseline or a percent of baseline.

## Statistical analysis

The number of neurons (n = 16 per group) and/or animals (nine control frogs and seven winter inactivity frogs) studied in synaptic and firing frequency experiments are consistent with other studies determining compensatory changes in neurophysiological properties in response to activity perturbations (e.g. *Wilhelm et al., 2009*; *Knogler et al., 2010*; *Lambo and Turrigiano, 2013*). For in vitro brainstem experiments of rhythmic respiratory activity in response to AMPA-receptor inhibition, four brainstems per group were used. As fictive respiratory burst amplitude has been shown to be insensitive to low concentrations of AMPA receptor inhibition (*Chen and Hedrick, 2008*), any changes in this insensitivity compared to control (i.e. deviations from no change) following the simulated winter environment would have been apparent in our 'before and after drug' experimental design. Similar sample sizes have been used previously for understanding synaptic transmission in the respiratory network from various species when the relative changes induced by receptor antagonists were large (*Greer et al., 1991*; *Johnson et al., 2002*). For comparisons of unpaired data that were normally distributed, we used a two-tailed, unpaired t test. If variance between the two groups differed, a Welch's correction was applied to the t test. Unpaired data that failed to meet the assumption of normality were compared using the Mann Whitney test on ranked data. Within preparation comparisons for recovery from DNQX were analyzed using a paired t test as this test has been shown to be robust for small sample sizes when the correlation coefficient is large (*de Winter, 2013*). For the mean F-I relationship of each group, we compared the mean slopes between each group with an analysis of covariance (ANCOVA). These analyses were performed using Graphpad Prism 6.01 (Graphpad Software, San Diego, CA). Cumulative distributions were compared with the Kolmogorov-Smirnov test using R (*R Development Core Team, 2014*) to compute exact p values. Detailed statistical information for each figure is provided in a table (*Supplementary file 1*). Error bars are presented as ±standard error of the mean unless otherwise specified. Statistical significance was accepted when p<0.05.

## Acknowledgements

We would like to thank the Department of Biological Sciences at Wright State University for the Biology Award for Research Excellence Grant and the Biomedical Sciences PhD program for stipend support (JS).

## Additional information

### Funding

| Funder | Grant reference number | Author |
| --- | --- | --- |
| Wright State University | Department of Biological Sciences: Biology Award for Research Excellence | Joseph M Santin |
| Wright State University | Biomedical Sciences PhD Program | Joseph M Santin |

The funders had no role in study design, data collection and interpretation, or the decision to submit the work for publication.

### Author contributions

Joseph M Santin, Conceptualization, Data curation, Formal analysis, Funding acquisition, Investigation, Methodology, Writing—original draft, Writing—review and editing; Mauricio Vallejo, Data curation, Formal analysis, Writing—review and editing; Lynn K Hartzler, Supervision, Investigation, Writing—review and editing

### Author ORCIDs

Joseph M Santin http://orcid.org/0000-0003-1308-623X
Lynn K Hartzler https://orcid.org/0000-0002-2355-0174

### Ethics

Animal experimentation: Experiments were approved by the Wright State University Institutional Animal Care and Use Committee (protocol number 1047).

### Decision letter and Author response

Decision letter https://doi.org/10.7554/eLife.30005.011
Author response https://doi.org/10.7554/eLife.30005.012

## Additional files

### Supplementary files

• Supplementary file 1. Statistical information for all analyses performed in the paper.
DOI: https://doi.org/10.7554/eLife.30005.009

• Transparent reporting form
DOI: https://doi.org/10.7554/eLife.30005.010

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
