## [Decision Letter]

Thank you for submitting your article "Synaptic up-scaling preserves motor circuit output after chronic, natural inactivity" for consideration by *eLife*. Your article has been reviewed by three peer reviewers, one of whom is a member of our Board of Reviewing Editors and the evaluation has been overseen by Eve Marder as the Senior Editor. The following individuals involved in review of your submission have agreed to reveal their identity: Carlos D Aizenman (Reviewer #2); Peter Wenner (Reviewer #3).

The reviewers have discussed the reviews with one another and the Reviewing Editor has drafted this decision to help you prepare a revised submission

Summary:

All three reviewers find this study, which reports of synaptic scaling in a respiratory circuit in an adult animal that is driven by a natural environmental stress, to be very interesting and worthy of publication in *eLife*, pending revisions. The scaling data are compelling, and demonstration of synaptic scaling in a behaviorally relevant circuit under perturbation that naturally occurs in nature is timely and important. A major revision is requested concerning the data presented in Figure 4, which involves addition of new text, re-analysis of data and inclusion of new plots. Based on the larger sensitivity of fictive respiratory burst responses to AMPAR blockade in wintered preparations compared to controls, the authors argue for the contribution of prominent AMPAR scaling in wintered animals. Nevertheless, the rationale for the design of experiments has not been clearly explained, and the "n" is rather small.

Essential revisions:

1) Please provide a clearer explanation for this set of experiments, taking into consideration that there could have been other reasons why motor neurons scaled up their synaptic AMPAR (instead of ensuring proper breathing when the respiratory demand increases in the spring). Moreover, aside from respiratory motor neurons, other respiratory circuit component(s) could have also undergone some form of compensation during wintering. Finally, there needs to be a more coherent discussion of how/why scaling contributes to the recovery of overwintering respiration (which is actually higher at baseline). Is there an overwintering-induced change in the circuit that prevents normal respiratory activity once temperatures are increased, and could scaling compensate for this issue?

2) One should include a comparison of raw data (not normalized) along with normalized data. This might clarify the presence of AMPAR-independent component of adaptation in wintered preparation.

3) Please explain/justify the use of small "n" number.

---

## [Author Response]

Essential revisions:1) Please provide a clearer explanation for this set of experiments, taking into consideration that there could have been other reasons why motor neurons scaled up their synaptic AMPAR (instead of ensuring proper breathing when the respiratory demand increases in the spring). Moreover, aside from respiratory motor neurons, other respiratory circuit component(s) could have also undergone some form of compensation during wintering. Finally, there needs to be a more coherent discussion of how/why scaling contributes to the recovery of overwintering respiration (which is actually higher at baseline). Is there an overwintering-induced change in the circuit that prevents normal respiratory activity once temperatures are increased, and could scaling compensate for this issue?

To the first point, the rationale, background, and hypothesis for this series of experiments have now been more adequately described. Although we agree that there could have been a number of reasons as to specifically why synaptic scaling occurred in these respiratory vagal motoneurons assessed in this study, the purpose of this series of experiments was to determine whether up-scaling of AMPA receptors had a functional impact, consistent with compensation, when these neurons received input from respiratory central pattern generator in the intact network. Using the in vitro brain-stem spinal cord preparations, we can record population motoneuron output through the vagus nerve associated with respiratory rhythmogenesis to provide a functional readout of both the respiratory CPG activity (i.e. fictive breath frequency) and motoneuron firing due to activation by the respiratory CPG (i.e. fictive breath amplitude). This allowed us to tie the changes in synaptic strength of respiratory vagal motoneurons identified in slices to a functional impact on network output by recording population output from these motoneurons during ongoing input from the respiratory CPG. This purpose has been more adequately described by rewriting this section of the paper and modifying the Figure 5 to make the relationship between the motoneurons in the slice and brainstem-spinal cord experiments more clear. New text for the rationale and description of the results from the series of experiments using in vitro brainstem-spinal cord preparations appears in the Results section.

To the second point, we agree that other aspects of the respiratory network, in addition to respiratory motoneurons, likely underwent compensation during overwintering disuse. Our data leads us to interpret that compensation occurs at (at least) two distinct loci, respiratory motoneurons and the respiratory rhythm generating network. Respiratory motoneurons (the focus of this study due to our ability to label them) use synaptic scaling to compensate for inactivity-related stimuli. Based on the experiments we performed we are confident that up-scaling of AMPA receptors on respiratory motoneurons (specifically of the vagal motor nucleus) appears to be compensatory in response to inactivity-related stimuli in the winter environment. This is based on evidence from respiratory motoneurons in slices where we observed upscaling and also with these same neurons in the intact network where we demonstrate the amplitude of their population activity during respiratory bursts decreases by ~40% after, presumably, blocking the effects of synaptic scaling with an AMPA receptor antagonist. Control bullfrogs do not alter burst amplitude upon exposure to the AMPA receptor antagonist at the same concentration. We suggested this is a compensation at the level of the motoneuron because in the absence of synaptic scaling burst amplitude would have been smaller had compensation not occurred during the winter. Mechanistically, this does not appear to be the case for rhythm generating circuits. We found here and previously (Santin and Hartzler, 2016) that fictive breath frequency is greater in frogs following overwintering. When assessing the change in burst frequency during application of the AMPA receptor blocker we showed that both overwintered and control frogs decreased fictive burst frequency by the same extent. Had the sensitivity to the AMPA receptor antagonist been greater after inactivity compared to before “winter” it is possible that this could have been interpreted as compensation in the respiratory CPG networks via AMPA receptors, but this did not occur. This implies that other mechanisms of compensation may be engaged during the winter to allow for a greater activity of the respiratory CPG following a winter’s worth of respiratory inactivity.

Our stance on this point has been made clear in the revision as we now separate these two possible loci of compensation (respiratory motor and respiratory CPG) in the Discussion section which details the interpretation of our data that leads us to conclude that scaling of AMPA receptors acts to boost (vagal) respiratory-related motoneuron output. The next paragraph details the notion that compensation likely occurs in the respiratory CPG network independent of AMPA receptor scaling. In this paragraph we speculate on a few non-mutually exclusive possibilities as the mechanisms that may underlie fictive lung breath frequency facilitation after winter inactivity. We feel this new text should provide “a more coherent discussion of how/why scaling contributes” to ventilation following overwintering inactivity of the respiratory motor output.

To the final point, we do not think our data support the idea that scaling “recovers” breathing in the spring, nor was this the message we intended to convey in the original manuscript. Our data support the idea that scaling of AMPA receptors on motoneurons provides the ability to generate a greater amount of motor drive compared to the drive that would be produced had synaptic scaling not occurred (hence the title “Synaptic up-scaling preserves motor circuit output after chronic, natural inactivity”). However, we cannot say whether scaling per se “recovers” or “initiates” ventilation after the winter. We think this is unlikely given that emergence and subsequent lung breathing may be more heavily driven by environmental cues associated with spring (e.g. ambient temperature). The present study addressed compensatory responses at the level of the motoneuron that seem to enable appropriate motor output to respiratory muscles after months of inactivity; however, how the CPG circuit “knows” how to start up again in the first place (e.g. mechanisms underlying the processing of information about metabolic state, ambient temperature, etc.) is an interesting and separate question.

In the “essential revisions” summary, the idea was posed that a circuit change inhibits ventilation immediately following emergence to initiate scaling for the recovery of ventilation. We think this is unlikely for a few reasons: (1) as soon as the frog is removed from the simulated overwintering environment, breathing movements start almost immediately (JMS personal observation and this has been noted in Santin & Hartzler, 2016). The information that bullfrogs resume breathing almost immediately has been added to the first paragraph of the Results section that details the characteristics of these frogs immediately after winter inactivity. (2) synaptic scaling during inactivity generally (but not exclusively) occurs slowly over longer time scales such as days when assessed in neurons from endothermic animals at physiological temperatures. Since we have previously demonstrated that no motor unit activity occurs in the glottal dilator muscle (i.e., the muscle innervated by the neurons we studied here that underwent AMPA receptor scaling) in the overwintering environment, there seems to be a two month time period in our study over which scaling in response to inactivity-related stimuli could slowly occur. Once the frog initiates breathing activity after the winter, we hypothesize that this would begin to provide the opposite signal (activity) to these respiratory motoneurons for the synaptic up-scaling we observed. (3) If increases in miniature EPSC were caused after the frog was removed from the winter environment (not during the “winter”), a rapid LTP-like mechanism might explain the larger mEPSC we measured. If this were true, the strongest synapses would strengthen the most, leading to an exponential relationship in the rank order mEPSC plot. However, a linear regression fits the data better than an exponential curve, implying AMPA receptor scaling rather than an LTP–like mechanism caused the increase in the mEPSC. We feel that these points collectively support a model where scaling of AMPA receptors on vagal respiratory motoneurons happens slowly (although we are not sure of the tempo of this process) and globally during the winter environment when respiratory motoneurons are inactive. Therefore, the amplitude of the “baseline” respiratory motor output and subsequent activation of respiratory muscles after overwintering inactivity likely represents, in part, the effects of scaling of AMPAR on these motoneurons. This has been made clear in the Discussion section in the revision where we describe the rationale for favoring a scaling mechanism over an LTP-like mechanism and, also in the Discussion section, where we now suggest that this process occurs during winter inactivity to support motor drive to respiratory muscles when breathing is again required. We hope this new text alleviates confusion.

2) One should include a comparison of raw data (not normalized) along with normalized data. This might clarify the presence of AMPAR-independent component of adaptation in wintered preparation.

Figure 5 (previously Figure 4) has been modified to now include all raw data points for motor burst amplitude, motor burst area, and fictive-breathing frequency before and after application of DNQX. For fictive breathing frequency, we have performed paired t-tests (before and after DNQX) and also we have compared absolute burst frequency at baseline between control and winter inactivity frogs using a Mann-Whitney test. This comparison of “baseline” burst frequency between the two groups replaced Supplemental Figure 1 in the first submission. Although we show raw data points for population motoneuron amplitude and area during respiratory bursts we did not perform a statistical analysis on the raw data because the signal generated by integrated population motoneuron firing is a “relative” measure. Similar to other extracellular recording techniques like electromyography, the amplitude of population motoneuron bursts recorded extracellularly through the cut nerve root of these rhythm generating respiratory preparations is only comparable to changes in the amplitude within a single preparation. Therefore, the individual values have no meaning in the absolute sense for use in statistical analysis. We only performed a statistical test for burst amplitude parameters when assessing changes (% of baseline) induced by DNQX as this is a valid calculation commonly used in the field to understanding respiratory transmission onto motoneurons in rhythm preparations (for example in neonatal rodents, Greer et al., 1992 and in adult turtles Johnson et al., 2002). The inability to compare absolute burst amplitudes in these preparations has been mentioned in the revised presentation of the results for the in vitro brainstem-spinal cord experiments (Results section). Although the data points certainly show the trends in the data, we want to be sure that readers do not interpret these values between preparations as having meaning in the absolute sense.

3) Please explain/justify the use of small "n" number.

The justification for our use of the number of in vitro brainstem-spinal cord preparations in Figure 5 (previously Figure 4) can be found in subsection “Statistical Analysis “of the revised manuscript. Per *eLife* standards, we also justify the sample size for the number of animals and neurons used in slice experiments (subsection “Statistical Analysis“). We concur that the number of in vitro brainstem-spinal cord preparations is indeed low; however, the effect that we were studying (changes in burst amplitude/frequency in response to bath DNQX application and differences between groups of frogs) was strikingly clear.

As the relative change in the amplitude of the population burst during application of antagonists of neurotransmitter receptors gives insight into that receptor’s role in respiratory CPG transmission to motoneurons, we were focused on identifying differences in amplitude sensitivity of respiratory motor bursts between control and winter bullfrogs in response to application of an AMPA receptor antagonist. There is a precedent in the literature that a low concentration of AMPA receptor blocker decreases the frequency of fictive-breaths, but does not influence the population respiratory motoneuron amplitude in control bullfrogs Chen and Hedrick, 2008). Thus, we chose a concentration of AMPA receptor blocker to recapitulate this effect in control frogs in our hands with the hypothesis that if scaling had a functional impact on respiratory motoneurons in the intact network, that respiratory transmission to these same neurons would be more sensitive to AMPA receptor block (i.e. a deviation from “no change”) when receiving rhythmic input from the respiratory CPG. In contrast to the control frogs, winter frogs had amplitudes that quite drastically differed from “no change” observed in control frogs. While a sample size of 4 preparations per group is small, it is relatively common when the magnitude of the effect of large (i.e. “presence or absence”). For example see references by Greer et al., 1992 and Johnson et al., 2002 that used between 4–6 preparations to draw “presence/absence” conclusions for synaptic transmission from the respiratory CPG onto respiratory motoneurons as we did here. In the spirit of full transparency we have provided integrated population respiratory motor bursts recorded through the vagus nerve that represent the mean from each control (top) and winter inactivity (bottom) bullfrog before (darker green) and after (lighter green) application of DNQX.

Additionally, we chose conservative non-parametric statistics to analyze these unpaired data as a sample size of 4 per group does not allow determination of whether these samples were drawn from a normal distribution (these are the p-values shown in Figure 5 of the revised paper). However, if we use conventional parametric statistics to analyze data of these sample sizes (which is fairly common in rhythmic respiratory preparations with low “n”), we achieve even lower p-values (p=0.01 for change in amplitude during DNQX and p=0.009 for burst area) compared to p=0.029 using a very conservative rank transformation for analysis of these data in the absence of information about normality.

In summary, we agree that our sample size is small; however, the consistency of the “presence vs. absence effect” we observed between these groups of frogs and its clear relevance to the point we are making (AMPA receptor scaling on individual respiratory motoneurons relates to increased reliance of AMPA receptors for transmitting the population motoneuron respiratory burst immediately after winter inactivity) leads us to question the benefit of sacrificing additional animals to add a few more frogs to each group.